

# WRF-Chem model simulations of a dust outbreak over the Central Mediterranean and comparison with multi-sensor desert dust observations

Umberto Rizza[1], Francesca Barnaba[2], Mario Marcello Miglietta[1], Gian Paolo Gobbi[2], Cristina Mangia[1], Luca Di Liberto[2], Davide Dionisi[2], Francesca Costabile[2], Fabio Grasso[1]

[1]CNR/ISAC, Unit of Lecce, Lecce, 73100, Italy

[2]CNR/ISAC, Unit of Roma, Roma, 00133, Italy

*Correspondence to*: Umberto Rizza (u.rizza@isac.cnr.it)

**Abstract.** In this study, the Weather Research and Forecasting (WRF) Model with online coupled chemistry (WRF-Chem) is applied to simulate an intense Saharan dust outbreak event that took place over the Mediterranean in May 2014. The dust outbreak was generated in correspondence with an omega-like pressure configuration associated with a cyclogenesis in the

Atlantic coasts of Spain. This pattern has been recognized as one of the three major cyclogenesis situations responsible for the transport of Saharan dust towards the Central and Western Mediterranean. In fact, in the case investigated here, a cyclone near the Atlantic coasts of Spain is responsible for strong westerly Atlantic winds (about 20 m s$^{-1}$) reaching the northern Sahara and leading to the lifting of mineral dust. The northward transport is made possible by a ridge over the central Mediterranean associated with the omega-like pressure configuration. WRF-Chem simulations are able to reproduce the synoptic

meteorological conditions and the transport outline of the dust outbreak that was in fact characterized by multiple, superimposed dust impulses. The model performances in reproducing the atmospheric desert dust load were evaluated using a multi-platform observational dataset of aerosol and desert dust properties, including optical properties from satellite and ground-based sun-photometers and lidars, plus in situ $PM_{10}$ data. This comparison allowed us to investigate the model ability in reproducing both the horizontal and the vertical displacement of the dust plume, and its evolution in time. Results show a

good agreement between the model and the AERONET-AOD in six sites in the Mediterranean. Comparison with the MODIS-AOD retrieval shows that WRF-Chem satisfactorily resolves the arrival, the time evolution and the horizontal pattern of the dust storm over Central Mediterranean. Comparison with lidar data confirms the desert dust advection to occur in several, superimposed 'pulses', as simulated by the model. In most cases the desert dust is shown to arrive above the PBL and then to descend and mix with the local aerosols within it. The vertical displacement of the dust was in good agreement with the lidar

soundings with a mean discrepancy along the aerosol extinction of about 40 – 60 %. The model-measurements comparison for the $PM_{10}$ and $PM_{2.5}$ shows a good temporal matching, although there is a clear overestimation of $PM_{10}$ and $PM_{2.5}$ of the order of 70 % during the dust peak. This tendency is reduced or even inverted in weak-dust or no-dust conditions, in which model and measured $PM_{10}$ and $PM_{2.5}$ are within 30 % and 10 – 60 %, respectively. For the $PM_{10}$ metrics it was also possible to



investigate the accordance between the model-based and the measurements-based dust-$PM_{10}$. This comparison confirmed the

$PM_{10}$ model overestimation to be related to over-predicted dust mass by a factor of 140 %.

**1 Introduction**

One of the main sources of uncertainty in our understanding of long-term climate variability is the role played by aerosols,

since the related uncertainty greatly exceeds that of the other mechanisms combined all together (Houghton et al., 2001; IPCC,

2007). When aerosol observations alone are analyzed, it is difficult to separate the influence of natural particles from the effects

of anthropogenic ones (Ghan and Schwartz, 2007). Among natural aerosols, mineral dust is the foremost specie, comprising

as much as 75 % of the global aerosol mass burden, as estimated by satellite products (Ginoux et al., 2012). Mineral dust

affects the earth's climate in many different ways, which are not completely understood and predictable. It influences the

atmosphere-Earth balance, directly by scattering and absorbing short- and long-wave radiation with consequences on the net

heating rates (e.g., Alpert et al. 1989 Balkanski et al., 2007). The uncertainties in the direct radiative forcing are primarily

attributed to the mineral aerosol shape (Kalashnikova and Sokolik 2002, Haapanala et al., 2012), but also on their optical

properties (Sokolik and Toon 1999) and their chemical composition (e.g., Claquin et al. 1998). In addition to these direct

effects, aerosol indirectly affects the radiative balance by modifying cloud properties (e.g., Rosenfeld et al., 2001, Ghan and

Schwartz, 2007, Karydis et al., 2011).

It has been estimated that about half of the global total natural dust emissions are generated in the Sahara Desert and its

surroundings (Goudie, 2009; Huneeus et al., 2011; Ginoux et al., 2012). Deep convection produced by the strong surface

heating can uplift mineral dust particles for several kilometers into the free troposphere, where they are finally advected over

large distances at the continental and intercontinental scales (Goudie and Middleton, 2001; Engelstaedter et al., 2006). A

considerable fraction of the dust loaded from Saharan sources remains in Africa, being transported and deposited in the

Sahelian countries along the well-known "meningitis belt" (Molesworth et al., 2003). Another significant fraction is

transported eastward across the Atlantic Ocean (e.g. Prospero and Mayol-Bracero, 2013; Yu et al, 2015), but large Saharan

dust amounts are also carried across the Mediterranean Sea to Europe (Moulin et al., 1998, Barnaba and Gobbi, 2004,

Israelevich et al., 2012) in episodic storms and/or following seasonal patterns (Barnaba and Gobbi, 2004, Pey et al., 2013).

During such outbreak events, mineral dust may be considered as the largest $PM_{10}$ source at urban and rural sites in the

Mediterranean basin (Kaskaoutis et al., 2012; Salvador et al., 2014; Pey et al., 2013, Kabatas et al., 2014), contributing to a

relevant percentage of the episodes of $PM_{10}$ daily limit exceedance (50 μg m$^{-3}$) registered at these sites (Salvador et al., 2014),

with peaks of contributions up to 80% of the total mass (Kaskaoutis et al., 2012).

During the year, the transport pathway of Saharan dust towards the Mediterranean is mainly determined by low-pressure

systems over the Atlantic or North Africa, high pressure over the Mediterranean region and/or high pressure at upper levels

over Africa (Moulin et al., 1998; Barkan and Alpert, 2008; Querol et al., 2009; Pey et al., 2013, Salvador et al., 2014). Using

Meteosat retrievals of dust optical depths, Moulin et al. (1998) showed that the northwards transport of dust follows a seasonal





pattern, being eastward when associated with the Sharav cyclones (Alpert and Ziv, 1989), and directed toward the western Mediterranean basin from March to August, caused by the coupling between a Saharan low and a Libyan high or by a cyclogenesis in the Atlantic coasts of Spain.

Modeling the transport of desert dust is receiving increasing attention from the scientific community, allowing to better

ascertain its impact on radiation budget (Hsu et al., 1999), clouds (Bangert et al., 2011), as well as on air quality and human health (Goudie and Middleton, 2001; Pey et al., 2013). Anyway, despite many improvements in characterizing dust source regions thanks to satellite products (Ginoux et al., 2012, Schepanski et al., 2012), modeling dust emission and transport is still challenging due to the high uncertainties associated to the diffuse character of the emissions, re-suspension processes, the inherent complexity of aerosol chemistry and meteorological conditions, which strongly influence dust outbreaks and their

spatial profiles (e.g., Knippertz, and Todd, 2012). This was evident in the intercomparison performed among 15 different global models in the framework of the Global Aerosol Model (AeroCom) initiative (Schulz et al., 2009) as well as in a recent intercomparison study among 9 European regional dust model simulations (Basart et al., 2016).

Aim of this study is to evaluate the capability of the WRF-Chem model to properly simulate an episode of mineral dust long-range transport occurred in the period 19-24 May 2014 over the Central Mediterranean regions. The event consisted of a series

of dust plumes generated in the Northwest Sahara by strong winds associated to an omega-like circulation, characterized by a low-pressure system localized in the Atlantic coasts of Spain. Dust plumes were transported northward, resulting in an intense (AOD at 550 nm > 1) dust event over the Mediterranean impacting mostly Italian and French sites on 21 and 22 May 2014. The event is well documented by the stations of the Aerosol Robotic Network (AERONET, Holben et al., 1998) located in the Mediterranean basin. The outline of the work is summarized in the following. The setup of the WRF-Chem model used here

is described in Section 2. Based on a previous sensitivity study (Rizza et al., 2016), in this work we used the dust emission scheme proposed by Shao (2001, hereinafter denoted as S01) that considers explicitly the two major emission mechanisms for mineral dust and a refined four-classes texture soil type, in which for each class a proper value for the soil plastic pressure (surface hardness) is considered. Data and methods used for the comparison with meteorological and aerosol fields are described in Sections 3. Results are shown and discussed in Section 4, in which we first assess the quality of WRF-Chem

model in reproducing the synoptic situation comparing the model prediction of the geopotential height with satellite data (4.1). By exploiting the multi-sensor dataset of aerosol and desert dust observations described in Section 2, in Sections 4.2 and 4.3 we evaluate the model performances in reproducing the dust outbreak in the horizontal and vertical scale, respectively. The former includes the identification of the main dust source regions for the addressed event. Concluding remarks appear in Section 5.

## 2 The Wrf-Chem model

The WRF/Chem is a fully coupled online community model for the prediction and simulation of weather, dispersion, air quality, and regional climate (Grell et al, 2005). The chemistry model has been built to be consistent with the Weather Research





and Forecasting (WRF, http://www2.mmm.ucar.edu/wrf/users/) modeling package. Possible applications of the current modeling system concern: the (i) prediction and simulation of weather, or regional and local climate; (ii) coupled weather prediction/dispersion model to simulate release and transport of constituents; (iii) coupled weather/dispersion/air quality model with full interaction of chemical species as well as particulate matter; (iv) study of processes that are important for global

climate change issues, including the aerosol direct and indirect forcing.

### 2.1 Model setup

In this study the WRF-Chem version 3.6.1 was used. Figure 1 shows the model domain, which covers North Africa, Southern Europe and the western part of Asia, with 160 x 90 grid points centered at lat = 30.6° N, lon = 18.7° E. In the same figure the

location of the AERONET stations that will be used in our following analysis is reported. The horizontal grid spacing is 50 km for both directions with 40 vertical levels up to 50 hPa. Boundary and initial conditions were extracted from NCAR/NCEP Final Analysis (FNL from GFS) (ds083.2), with 1-degree resolution, every 6 hours.

#### *2.1.1 Physical parameterizations*

As shown in table 1, the following physical schemes are used: the Mellor–Yamada–Nakanishi and Niino (MYNN) 2.5 level turbulent kinetic energy (TKE) parameterization for the planetary boundary layer (Nakanishi et al., 2009); the MM5 similarity scheme (Paulson, 1970) and the RUC Land Surface Model (Benjamin et al., 2004) are chosen to represent the surface layer physics and the land surface parameterization. The Rapid Radiative Transfer Model (RRTMG) for both short-wave (ra_sw_physics = 4) and long-wave (ra_lw_physics = 4) radiation is used for the aerosol direct radiative effect (Mlawer et al.,

1997). The Purdue Lin scheme (mp_physics = 2) is used for the treatment of the microphysics processes, with all parameterization production terms based on Lin et al. (1983) and Rutledge and Hobbs (1984) with some modifications, including saturation adjustment following Tao et al. (1989) and ice sedimentation. This setup is compatible with the shortwave radiative feedbacks (or what is known as the direct effect), which are included with the running of chemistry. Concerning the aerosol optical properties, the Maxwell-Garnett mixing rule (aer_op_opt = 2) is adopted in its approximate parameterization,

that is considering small spherical randomly distributed black carbon cores in particle (Bohren and Huffman, 1983). For the dry removal of dust aerosols, a dry deposition scheme including gravitational settling and surface deposition is used (Wesely, 1989). A wet deposition scheme that consider rainout/washout in large-scale precipitation (Balkanski et al., 1993) is used for both sea-spray and dust aerosols (Su and Fung, 2015).

#### *2.1.2 GOCART aerosol module*

As aerosol/chemistry module, the GOCART scheme (Giorgia Tech/Goddard Global Ozone Chemistry Aerosol Radiation and Transport model, Chin et al., 2000) was selected (*chem_opt=GOCART_SIMPLE*). It produces output for 7 bulk aerosol species, organic carbon (OC1, OC2), black carbon (BC1, BC2), other GOCART primary ($PM_{2.5}$, $PM_{10}$) and Sulfate (only





secondary aerosol species), and for 8 sectional aerosols species: 4 dust bins (0-2.5, 2.5-5, 5-10, 10-20 μm) and 4 sea salt bins (0.3, 1.0, 3.2, 7.5 μm). GOCART comes with simple sulfur gas phase chemistry including Dimethyl Sulfide (DMS) and sulphur dioxide ($SO_2$). While dust and sea-salt emissions are surface wind speed dependent, the others are prescribed from emissions inventories. In this context, the 3-dimensional background fields for OH, $H_2O_2$, and $NO_3$, 2-dimensional background fields for

dimethyl sulfide (DMS), as well as emissions fields for organic carbon (OC), black carbon (BC), sulfur dioxide ($SO_2$), and particulate matter (PM), are obtained from the PREP-CHEM-SRC emission preprocessor package (Freitas et al., 2011). This preprocessor reads the global antropogenic emissions from the RETRO reanalysis (http://retro.enes.org) and EDGAR (http://edgar.jrc.ec.europa.eu) emission database and the GOCART static background fields. These fields are provided to the program convert_emission (included in WRF-Chem public release) to produce the gridded netCDF emission files for the WRF-

Chem domain.

### 2.1.3 Dust emission parameterization

In any dust emission model, the basic parameters are: the threshold friction velocity at which dust particles begin to move, the horizontal and the vertical sand dust flux. The emission of dust particles may be classified considering the wind conditions at

the surface. In particular, under strong wind conditions the surface wind-shear is the principal dynamic parameter and the dust emission is generally a function of the threshold friction velocity. Under these conditions two main dust emission mechanisms have been recognized: saltation bombardment (Marticorena and Bergametti, 1995) and aggregate disintegration (Shao, 2001). Another important mechanism, when the lower troposphere is in the free-convective regime, is the direct aerodynamic lifting (Klose and Shao, 2012).

The model WRF-Chem version 3.6.1 includes three packages for mineral dust emission, the first two (dustgocart and dustgocartafwa) from the GOCART model, the third (dustuoc) from the University of Cologne group (Shao 2001, 2004, 2011). A preliminary comparison has been performed between the GOCART/AFWA and Shao 2001 (S01) schemes, and the result are discussed in Rizza et al. (2016). These show that the emission schemes based on GOCART/AFWA model produce a relevant over-prediction of dust concentration. This may in part be explained by the fact that the AFWA scheme considers

vertical dust flux to be related only to the clay content while the S01 scheme considers a more realistic soil texture type. For these reasons, in this work we have utilized the S01 scheme, which will be briefly described below.

Shao (2001) proposed a dust emission parameterization that considers explicitly the two major dust emission mechanisms described above. In particular, the aggregate disintegration is modeled with the hypothesis that dust aggregates fragment as they hit the surface. He proposed a size-resolved dust emission equation by supposing that particles are divided into $n$ particle-

size intervals and expressed the total dust flux ($F$) as an integral of the dust emission rate for particles of size $d_i$ by saltation of particles of size $d_s$:

$$F = \int_{d_i}^{d_s} \int_0^{d_i} \tilde{f}(d_i, d_s) p(d_i) p(d_s) \delta d_i \delta d_s$$





where $\tilde{f}$ is the dust emission rate for particles of size $d_i$ generated by the saltation of particles of size $d_s$, and $p(d_{i,s})$ can be regarded as a combination of two idealized particle size distributions, known as minimally disturbed particle size distribution $p_m(d_{i,s})$, and fully disturbed particle size distribution $p_f(d_{i,s})$ whose values for each soil category are provided in a look-up table. As $p_m$ and $p_f$ are functions of land surface properties, the soil data used in WRF-Chem play an important role in dust emission

simulation. In this study, the default soil categorization data set from the United States Geological Survey (USGS) with 5′ geographic resolutions was selected. The effective soil texture is obtained from the USGS 12 classes considering only 4 types classes: sand, sandy clay loam, loam and clay. More details of this formulation may be found in Shao (2001), Kang et al. (2011) and Su and Fung (2015). Chemistry and dust emissions parameterizations adopted here are reported in Table 2.

### 3 Observational Dataset

**3.1 Meteorological fields from AIRS**

The analysis of the geopotential at 500, 700 and 850 hPa, and the water vapor mixing ratio at 700 hPa, is accomplished using the NASA-Aqua Atmospheric Infrared Sounder (AIRS) Level 3 daily standard physical retrieval V006 gridded products at 1 x 1 degree. The time averaged map of geopotential and water vapor mixing ratio are obtained combining either the descending (equatorial crossing North to South @1:30 AM local time) and the ascending (equatorial crossing South to North @1:30 PM

local time) orbit.

**3.2 Aerosol horizontal field**

The observation-based characterization of the aerosol field over the horizontal scale is made here by using both a network of sunphotometers located at multiple sites and measuring synchronously, and satellite retrievals capturing wider areas within a

single passage. In particular, sunphotometers at Central Mediterranean sites operating within the worldwide AERONET aerosol robotic network are used to evaluate the columnar aerosol optical properties over the investigated area. These measurements are complemented by the aerosol retrievals from the MODIS sensors on board the NASA platforms Terra and Aqua.

*3.2.1 AERONET AOD dataset*

AERONET is a federation of ground-based sunphotometers established by the US NASA and currently led by NASA and the French CNRS. It includes nearly 1000 sunphotometers spread worldwide, whose data are processed following the same aerosol retrieval procedures (Dubovik et al, 2000a, Dubovik et al, 2000b, Dubovik et al, 2006) and made available in quasi real-time through the NASA portal aeronet.gsfc.nasa.gov.

The main quantity measured by sunphotometers is the aerosol optical depth (AOD), an optical parameter quantifying the aerosol load in the whole atmospheric column. The AOD is unit-less, as it can also be obtained as the integral over altitude of the aerosol extinction coefficient (units of length[-1]).





In this study we use Level 2 (L2, i.e. cloud screened and quality assured) AOD measurements in the visible spectrum performed at those stations localized in the Central Mediterranean fulfilling the following requirements: (i) consistence with the spatial pattern of the dust intrusion, i.e. location close to the area affected by the investigated dust outbreak; (ii) availability of L2 data in the period considered. The resulting six stations are shown in Figure 1 and include the AERONET sites of Ersa (Corsica,

France, lon = 9.359° E lat = 43.004° N, elev = 80 m), Calern (France, long = 6.927° E, lat = 43.749° N, elev = 1270 m), Carloforte (Sardinia, Italy, lon = 8.310° E, lat = 39.140° N, elev = 15 m), Rome (Italy, long = 12.647° E, lat=41.840° N, elev = 130 m), Modena (Italy, long=10.945° E, lat=44.632° N, elev = 56 m) and Ispra (Italy, long = 8.627° E, lat = 45.803° N, elev = 235m). Location of these sites is shown in Figure 1. The uncertainty in AOD measurements from CIMEL field instruments is mainly due to calibration uncertainty. Eck et al. (1999) estimated this to be ~ 0.01 in the visible and near-IR, increasing to

~ 0.02 in the ultraviolet.

### 3.2.2 MODIS AOD dataset

The MODerate-resolution Imaging Spectroradiometer (MODIS, Salomonsen 1989) instrument flies on board the NASA TERRA spacecraft since December 1999. It has 36 wavelength bands spanning from the visible to the infrared, high spatial resolution, and near daily global coverage, overpassing the equator at 1030 LT. Aerosol characterization is a core MODIS

mission (Kaufman et al., 1997) and the AOD is still the most robust aerosol physical parameter derived from space. In particular, two different approaches are used to retrieve the AOD from MODIS data, referred to as 'Dark Target' (DT, Kaufman et al. 1997) and 'Deep Blue' (DB, Hsu et al., 2004). The algorithm at the basis of the 'Dark Target' approach is further differentiated when applied over ocean (Remer et al., 2005) or land (Levy et al. 2007a, b) surfaces, and it is not suitable to be applied over bright surfaces (deserts, snow). The Deep Blue approach was developed to fill this gap (Hsu et al., 2004) and well

complements the 'Dark Target' retrievals. The most recent collection 6 (C006) of MODIS AOD data provides a single AOD product combining both the DT and the DB AOD retrievals and, as considered the "best-of" product for most quantitative purposes (Levy et al., 2013), it was used in our study (in particular we use here the MODIS daily product MOD08_D3 v6).

### 3.3 Aerosol altitude-resolved view over Rome (Italy)

The characterization of the aerosol field over the vertical scale is made here by employing continuous (h24) lidar/ceilometer measurements performed in one of the six AERONET sites, i.e., the Rome - Tor Vergata site. Even if this represents a single observational point, it lies just in the middle of the area investigated, and is therefore expected to be particularly suitable to evaluate the model capability to reproduce the dust plume vertical extent and its transport timing.

The lidar-ceilometer measurements were performed in the framework of the EC-LIFE+ project DIAPASON (Desert-dust

impact on air quality through model-predictions and advanced sensors observations, www.diapason-life.eu). More details on the Project and relevant results can be found in Gobbi et al. (2016) and in Barnaba et al. (2016), while the instruments characteristics and relevant dataset used here are described hereafter.





### 3.3.1 Lidar datasets

The aerosol vertical profiles used in this study were collected by two different co-located instruments: a commercial CHM15K ceilometer (Lufft Mess und Regeltechnik GmbH, www.lufft.com), and a research-type lidar (ATLAS) developed at the ISAC-CNR laboratories. The former is now part of an in-progress Italian network of such systems (the Automated Lidar-Ceilometer

Network, ALICENET, www.alice-net.eu), which are conversely already widely employed in Germany, where the national meteorological service (DWD) operates over 50 of these instruments (e.g., Flentje et al. 2010a, Wiegner and Geiß, 2012). In 2010 the DWD ceilometer network was able to track the spatio–temporal evolution of the Eyjafjallajökull eruption volcanic plume (Flentje et al., 2010b).

The CHM15K instrument uses a pulsed Nd:Yag laser source at 1064nm with an output laser energy of about 8 μJ, a pulse

repetition rate of 5–7 kHz and a vertical resolution of 15 m. Its configuration allows to sound the aerosol load in the atmosphere in the range 150 m – 15 km.

As for all lidar systems, the signal in the lowermost atmospheric levels has to be corrected due to the incomplete superposition of the laser and the receiver field-of view (FOV). For this system the overlapping correction function is provided by the manufacturer, which determines the correction in the factory using a reference instrument.

The ATLAS system is a further miniaturization of a previous mobile, polarization-sensitive lidar system (VELIS) developed by ISAC-CNR (Gobbi et al., 2000). ATLAS maintains most of the VELIS characteristics although it uses a different 1 kHz, 30 μJ/pulse laser source and reaches full overlap at approximately 500 m. As VELIS, ATLAS has two receiving channels, collecting respectively the light backscattered by particles in the parallel and polarized plane with respect to the laser emitted one. Since spherical particles do not change the polarization plane of the incident light while non-spherical particles do, the

comparison of the two lidar channels allows to detect presence of non-spherical, e.g. of mineral particles, in the atmosphere (e.g., Gobbi 1998). An example of this capability is provided in Section 4.3.

Both the CHM15K and ATLAS are able to work unattended in continuous mode (h24), and their measurements are therefore used here to investigate the capability of the model to reproduce the dust plumes over Rome in terms of both temporal matching and vertical extent (Section 4.3).

In this work we use both the range-corrected signal of both systems (RCS, to a first approximation related to the aerosol amounts) and some (more quantitative) aerosol extinction profiles from ATLAS (see Section 4.3). In fact, the inversion of the lidar RCS into aerosol optical properties requires the employment of the backward solution of the Klett inversion algorithm (Klett, 1981) to the data. In addition to the estimation of the molecular backscatter and extinction coefficients ($\beta_m$ and $\alpha_m$, respectively, calculated from climatological monthly air density profiles), the solution requires two assumptions: a boundary

value at a reference height $z_0$ where $\beta_a(z_0) = 0$ (Rayleigh calibration) and a so called 'lidar ratio' ($S_a = \alpha_a/\beta_a$). In our case, a calibration constant was derived applying the Rayleigh calibration to nighttime and cloud-free signals averaged over 1 h at 75 m height resolutions. For the second assumption, $\alpha_a$ is computed using a functional relationship $\alpha_a = \alpha_a(\beta_a)$ derived by Barnaba and Gobbi (2001) assuming non-spherical desert dust particles. The expected error on $\alpha_a$ is of the order of 30 %.



This approach requires an iterative inversion technique to correct the backscatter signal for extinction losses until convergence in the integrated aerosol backscatter (IAB=$\sum_0^{zcal}\beta_a(z)$) is reached. The estimation of the aerosol extinction coefficient below complete superposition of the laser and telescope FOV is obtained from a linear fit of the first two valid lidar points.

### *3.3.2 In situ PM$_{10}$ data*

To complement the column-integrated and the vertically-resolved aerosol optical properties described above, for the Rome site the observational dataset used to test the model also includes the standard particulate matter (PM) metrics regulated by the EU Air Quality legislation (i.e., the daily average PM$_{10}$ and PM$_{2.5}$ data) plus hourly resolved measurements at the Rome–Castel di Guido site (about 15 km W of the City Center). These data were collected using a SWAM dual channel instrument (FAI,

Italy), providing mass concentration measurement on hourly basis thanks to as specific application of ß technology including information about atmospheric mixing ratio (http://www.fai-instruments.com/images/img/pdf%20eng/DOC401.PDF). Relevant results are provided in Section 4.3.1.

### 4 Results and Discussion

### 4.1 Model capability to correctly reproduce the meteorology driving and associated to the dust event

Several authors evidenced that the northwards dust transport pathway from Sahara follows a seasonal pattern, changing from the eastern to the western Mediterranean basin during spring and summer (Moulin et al., 1998; Barnaba and Gobbi, 2004; Engelstaedter et al, 2006).

As a case study representative of the conditions occurring in spring, we selected an intense dust episode affecting the Central Mediterranean between 19 and 24 May 2014. This case corresponds to one of three different major cyclogenesis situations

that are thought to be responsible for the northwards transport of Saharan dust toward the Mediterranean (e.g., Engelstaedter et al., 2006), that is the cyclogenesis in the Atlantic coasts of Spain.

The synoptic analysis of the dust event is described using the AIRX3STD satellite products and not the usual reanalysis products. In Figure 2 we show the geopotential at upper (500 hPa) and lower (850 hPa) levels averaged in different sub-periods within 16-25 May 2014. The 500 hPa geopotential height maps show for the first three days (16-17-18) an intense zonal flow

in the southern Mediterranean (Figure 2a) as a consequence of a pressure low centered over the Balkan area and a high pressure over northeastern Africa. The following days (19 to 24) are characterized by an omega-like circulation, which is consequence of the northward expansion of the ridge toward the central Mediterranean and of the intensification of a pressure minimum over Spain, that brings (i) strong westerly winds in the northern Sahara and (ii) south-westerly flow over the western Mediterranean (Figure 2b). During the last simulated day (25 May, not shown) we have a further rotation of wind, which blows

from west-south-west direction.

At lower levels (850 hPa), AIRX3STD retrievals for the period 16-22 May show the presence of a high pressure over Libya and Egypt and of a low over Spain and Morocco that intensifies the southerly wind in the western and central Mediterranean (Figure 2c). From May 22 to 24, the low pressure moves eastward and northward, producing a rotation of the low level wind



from west over the western Mediterranean (Figure 2d). At the end of the period, a cyclonic circulation prevails over the Mediterranean, but with weak winds over most of the basin. A high pressure of limited extension from Libya to southern Italy determines southerly currents confined over the southern Mediterranean.

The ability of the model to reproduce the meteorology driving/associated with the dust event is evaluated in terms of
geopotential height (Figure 3) and water vapor mixing ratio during specific days (Figure 4). Both fields are obtained from AIRS retrievals at 700 hPa (panels a, c, e, in Figures 3 and 4) and compared to the WRF-Chem simulations (panels b, d, f) for the selected dates (21, 22 and 23 May). These two quantities are important because (i) the geopotential at 700 hPa gives an indication of the circulation pattern associated with the dust transport in the low-middle troposphere and (ii) the water vapor strongly influences the aerosol chemistry.

Figure 3a shows a low pressure over the Atlantic coasts of Spain on May 21 and a ridge extending from northern Africa to the central Mediterranean. Together with a low over Turkey, the whole pattern resembles an omega-like configuration, responsible for southwesterly wind over western Mediterranean. This circulation transports very humid air toward the Mediterranean basin from the tropical regions, and is delimited on its northeastern side by dry air centered over the central Mediterranean (Figures 4a). Another area of relatively high moisture can be found in the eastern Mediterranean, associated with the wide cyclonic
circulation near Turkey.

On May 22, the ridge is still persistent over the Italian Peninsula and responsible for the southwesterly wind over the western Mediterranean, which brings air of African origin toward northern and central Italy and southern France (Figure 3c). As a consequence, air with a high water vapor mixing ratio enters the Mediterranean from the south, increasing the humidity content up to northern Italy (Figure 4c). On May 23, the progressive weakening of the ridge (that at the end of the period is confined
over the southern part of the central Mediterranean) and the northward movement of the low over western Europe produce more zonal (west-south-westerly) currents over the western Mediterranean (Figure 3e). As a consequence, the moist tongue over the central Mediterranean is advected eastward, while drier air reaches the western part of the basin from the Atlantic Ocean (Figure 4e). Figures 3b, d, f show that the synoptic conditions and the wind field components (u, v) at 700 hPa are well reproduced by the WRF-Chem simulation for each of three selected days considered here (21, 22, 23 May). Although the
simulated and observed patterns of water vapor mixing ratio are qualitatively similar, there are some differences in their intensity and extensions, as the model generally overestimates the mixing ratio transport from Africa (Figure 4).

**4.2 Model capability to reproduce the horizontal pattern**

*4.2.1 - Identification of desert dust source areas*

As described above, an important condition for the existence of a dust source is the availability of fine grained material, which can be lifted from the ground when the surface wind speed exceeds a definite threshold. The threshold wind velocity depends on the surface roughness and grain soil size and in literature it is found to vary from about 6 to 9 ms$^{-1}$ (e.g., Chomette et al. (2006) report values ranging from $6.63 \pm 0.67$ to $9.08 \pm 1.08$ m s$^{-1}$).





In order to evaluate the location of the dust sources that are directly connected with the investigated dust intrusion, in Figure 5 we superimpose the modeled AOD at 550 nm (shaded contours) and the total dust flux (white contours for the selected dates of May 18, 20, 21 and 24 (panels a, b, c, and d respectively). The AOD is obtained from WRF-Chem simulations vertically integrating (from the ground to the top of domain, i.e. 20 km) the aerosol extinction coefficient at 550 nm. The same figures

also show: i) the wind field at 10 m (black arrows), that is directly connected with the dust emission, and ii) the wind field at 700 hPa (white arrows) connected with the long range transport of dust.

The model results show that four major dust plumes were generated in different source regions of Northern Sahara on May 18, 20, 21 and 24 respectively. Then, these dust plumes were transported toward the western Mediterranean and were responsible for the consequent AOD peaks registered by AERONET stations and of the aerosol vertical distribution observed

by lidar in Rome (Italy) (see also Section 4.3).

The first dust plume (Figure 5a) was generated by easterly surface winds of approximately 20 m s$^{-1}$ (black arrows) in the region marked by the ellipsoid S1. The peak value of about 100 μg m$^{-2}$ s$^{-1}$ was located at about (lat, lon) =(34° N, 8° W) that roughly correspond to the source area of Chott el Jerïd in Tunisia (Ginoux et al., 2012). This is a large endorheic Salt Lake (Chott), which become salt flats as it dries. The 700 hPa wind (white arrows) shows that the dust was transported toward the western

Mediterranean. The emission took place between 0600 UTC and 1600 UTC on May 18. To give a reference for the following analysis, the model-based temporal evolution predicts this dust plume originated in S1 to reach Rome approximately at 0800 UTC on May 19 at above 2 km altitude.

The second dust plume (Figure 5b) was generated by westerly surface winds of approximately 20 m s$^{-1}$ (black arrows) in the region marked by the ellipsoid S2. The peak value of about 100 μg m$^{-2}$ s$^{-1}$ was located at about (lat, lon) =(30° N, 5° W) that

roughly correspond to the source region of the Grand Erg Occidental Desert (Ginoux et al., 2012). The 700 hPa wind (white arrows) shows that the dust was transported toward the western Mediterranean. The emission took place between 1200 UTC and 1800 UTC on May 20. According to the model simulations, the core of this second dust plume reached Rome around 2200 UTC on May 21, at above 3 km altitude.

The third and most intense dust plume was generated on May 21, between 1000 UTC and 1800 UTC. The source region is

shown in Figure 5c and it is delimited by the ellipsoid S3 whose peak value of roughly 120 μg m$^{-2}$ s$^{-1}$ is located at (lat, lon)=(33° N, 0) corresponding to the area of Chott ech Chergui in the northwestern Algeria (Ginoux et al., 2012). The surface wind was almost 20 m s$^{-1}$, while the wind speed at 700 hPa (white arrows) again shows a northeastward transport. For this third plume the emission took place between 1000 UTC and 1800 UTC on May 21. It traveled very fast and arrived over Rome around 0400 UTC on May 22. A fourth, weaker dust impulse is produced on May 24. It was generated on May 24 at 1400 UTC on

the region delimited by the ellipsoid S4 in Figure 5d and then it reached Rome in the evening of the same day.

All these four source areas (S1, S2, S3 and S4) are located within a persistently active source region that is situated south of Atlas Mountains and are characterized by a system of ephemeral salt lakes that stays dry in summer, but receives some water



in winter. In this way they may play an important role in modulating dust emissions (Engelstaedter et al., 2006; Salvador et al., 2014).

### 4.2.2 Comparison of WRF-Chem and MODIS AOD over the Mediterranean

Figure 6 shows the MODIS-Terra combined Dark Target (DT) and Deep Blue (DB) AOD at 550 nm (MOD08_D3 collection 6 products) for the period 20-23 May 2016. The DB algorithm is only available over land, while the DT algorithm derives aerosol properties over ocean and land (excluding bright surfaces as deserts). The corresponding modeled AOD is given in Figure 7, which includes also the wind field components (u,v) at 700 hPa. As the Terra platform overpasses the equator at 1030 LT, the model results at 1100 LT have been considered for comparison with MODIS retrievals. The analysis of May 20

shows that the model-AOD and MODIS-AOD have a coherent spatial pattern. In MODIS data (Figure 6a) the highest AOD is located over the coasts of Libya and to the south of Tunisia, while the model-AOD (Figure 7a) roughly indicates a dust transport toward that region, with the peaks located mainly inland and shifted more to the west compared to the observations. Both MODIS and model data do not show a marked dust transport toward the Mediterranean yet. The first intrusion in the western Mediterranean is evident the day after, when the satellite data (Figure 6b) show a deep and intense dust frontal region extending

from the coasts of Algeria and Tunisia to southern France. WRF-Chem AOD (Figure 7b) shows a similar spatial distribution of the dust front, associated with southwesterly winds at 700 hPa. The model transport toward the Mediterranean appears slightly delayed, since high values of AOD are still present in the Sahara, differently from the observations. The comparison between model and observations for 22 and 23 May is made difficult because of the extensive cloud coverage in the analyzed region, which prevents the AOD retrieval from space (Figures 6c,d). For these two days WRF-Chem shows the dust outbreak

toward Sardinia/Corsica and northern Italy on May 22 (Figure 7c) and toward central-southern Italy and the Balkans on May 23 (Figure 7d), i.e., in the regions mostly covered by clouds, as frequent in dust-load conditions. The wind fields at 700 hPa show a progressive intensification of zonal flows.

Overall, the picture that emerges from the simulations is that of a strong intrusion of Saharan dust in the Mediterranean basin starting from May 21. An extensive dust front is formed and transported northward carried out by southerly winds at 850 hPa

and southwesterly currents at 700 hPa. A second dust plume enters the Mediterranean late on May 21 and is transported toward Sardinia/Corsica and the Tyrrhenian Sea during May 22. In the following hours the prevailing zonal flow prevents from further intrusion of Saharan dust into the basin.

### 4.2.3 Comparison of WRF-Chem and ground-based AOD at specific AERONET Mediterranean sites

To complement the comparison with the satellite observations and better follow the temporal evolution of the AOD field over the Central Mediterranean, we used the AERONET measurements at the six sites shown in Figure 1. Figure 8(a-f) depicts the hourly-resolved AOD at 550 nm from the WRF-Chem simulation (continuous red line) in comparison with the AERONET measurements in the six stations (black squares). The AERONET measurements show the highest AOD peaks occurred on



May 21 at Carloforte (Figure 8a), Sardinia (AOD = 1.2), and on May 22 at Ersa (Figure 8c), Corsica (AOD = 1.8), with high

values also measured in Central (Figure 8b) and Northern (Figure 8d) Italy (Rome, up to AOD = 0.75 and Modena up to AOD

= 0.85). Note that, due to the extensive cloud coverage over the Tyrrhenian Sea, Carloforte has no data on 22 May and Rome

on 23 May. The other two stations, Calern (Figure 8e) in Southern France and Ispra (Figure 8f) in Northwestern Italy, have

values of AOD less than 0.4 for all the simulated period. At Rome and Modena, WRF-Chem shows similar time evolutions

and generally reproduces pretty well the AERONET measurements. At Carloforte station, the simulation shows three AOD

peaks on 19, 21 and 22 May, while from the measured data only the peak on 21 May comes out, since in the other two days

there are no measurements due to the extensive cloud coverage. At Ersa station, the simulation shows a main peak on 22 May,

which is in agreement with the observations; in general, the measured AOD is well reproduced for the whole period. The

simulation in the other two sites (Calern and Ispra) shows a time pattern similar to that from measurements, with AOD values

generally lower than 0.4, indicating a minor impact of the dust plumes at these locations.

This analysis of AERONET data and its comparison with simulations shows that the first dust intrusion occurred on 21 May

up to southern Sardinia, the second and most intense dust plume occurred on 22 May, penetrating up to northern Corsica and

central Italy. This result confirms and integrates the analysis of satellite data.

### 4.3 Model capability to reproduce the desert dust plume vertical patterns;

The vertical evolution of the desert dust plume over Rome is revealed by the lidar and ceilometer measurements (see Section

2). Figure 9 shows the altitude (0-6 km) versus time (h24) cross section of the aerosol field as detected by the ceilometer

CHM15k (Figure 9a) and by the ATLAS  lidar systems (Figure 9b) for the period 19-25 May 2014. In particular, Figure 9a

shows the logarithm of the range-corrected (RC) CHM15k elastic signal (S) (i.e., $\ln(RCS) = \ln(S \times R^2)$), which, to a first

approximation, is proportional to the amount of particles loaded into the atmosphere, while Figure 9b shows the particles

depolarization ratio, obtained from the two ATLAS receiving channels. This parameter is a marker for non-spherical particles,

and therefore is particularly suitable to follow the desert dust plume evolution in space (vertical scale) and time.

The ceilometer measurements (Figure 9a) also show well the aerosol-tracked planetary boundary layer (PBL) development in

each day of the period considered. This can be identified by the green, bell-shaped area reaching a maximum altitude of about

2 km (particularly evident on May 20, 21, 22). On May 19 and 23, the boundary layer signal is somehow 'perturbed' by the

presence of rain and clouds (in red in Figure 9a).

Above the PBL, elevated aerosol layers are also clearly visible in the ceilometer trace; these have been highlighted by dotted,

white oval shapes in Figure 9a. Although the elastic ceilometer signal of Figure 9a does not discriminate the aerosol type, these

30    elevated layers over Rome are typically associated to Saharan dust (e.g. Gobbi et al., 2004, Gobbi et al., 2013). To prove these

layers are actually composed of mineral (non-spherical) particles, Figure 9b shows them to produce a typical depolarization

signal (Volume depolarization > 8%). To facilitate the comparison with the model outcome, in Figure 9c the same dust-

identification shapes have been superimposed on the relevant model-derived cross sections.





Overall, the lidar measurements in the period 19-25 May show the desert dust advection to occur in several, superimposed 'pulses', thus confirming the 'pulsed' nature of this event. In most of the cases the desert dust is shown to arrive above the PBL and then to descend and mix with the local aerosols within it. In particular, the four main different plumes, already discussed in Section 4.2.1, can be identified in the lidar/ceilometers records. A first plume arrives over Rome on May 19 (the

presence of clouds prevents from observing the exact time of its arrival over Rome); then, it progressively descends and is firstly detected at the ground on May 20. This is likely the plume originated in the area identified by S1 in Figure 5. A second plume (the one likely originated in S2, see Figure 5) arrives aloft in the evening of May 21 and then descends towards the ground. This plume superimposes to the previous one, and to a third major plume arriving in the afternoon of May 22 and extending from the ground up to 6 km (this is the major plume the model identifies to originate in the source region S3). The

mixing of the three plumes is observed down to the ground until at least May 24. As predicted by the model, a fourth and weaker pulse arrives aloft in the night between 24 and 25 May and superimposes to the previous two until the end of the addressed period.

Although qualitatively, lidar measurements at high vertical and temporal resolution allow to evaluate the model capability to reproduce the desert dust plume vertical patterns and timing. In particular, the comparison of Figures 9a,b with Figure 9c

shows the model to well reproduce the 'pulsed' pattern of this desert dust advection, reproducing pretty well both its timing and vertical extent. Some difference is found with the timing and vertical location of the fourth plume, which the measurements indicate to arrive at about 2 km around noon of May 24 and the model predicts at lower altitudes and with some hours of delay. Nevertheless, the model is still capable to reproduce the second peak of this fourth plume observed by the lidar systems aloft (above 3 km).

A more quantitative validation of the vertically-resolved model output is provided in Figure 10, in which the ATLAS lidar range-corrected signal (RCS) is inverted in terms of aerosol extinction coefficient (see Section 3.3.1). For this purpose, night-time/early-morning profiles have been selected to improve the signal-to-noise ratio in the measurements and thus facilitate the lidar signal inversion. Figure 10 shows that the model mostly reproduces the vertical pattern, with a double layer structure clearly evident on May 22 (Figure 10a, b). The elevated layer is likely uniquely composed of desert dust particles (as revealed

by the lidar depolarization trace in Figure 9), while in the PBL aerosol layer, desert dust is mixed with particles of local origin. Overall, for this date, which correspond to the maximum desert dust load over Rome (see Figure 8b), the model reproduces rather well the associated aerosol extinction along the vertical profile, with an estimated Normalized Mean Bias (NMB) of ±50% with respect to the lidar. This provides further insight in the good matching between the model and the measured AOD over Rome on May 22 (Figure 8b). Similarly, the vertically-resolved comparison for May 25 (Figure 10c, d) allows to better

understand the model AOD underestimation (Figure 8b) in such lower desert dust loads. In fact, although the model is still able to reproduce the shape of the aerosol profile, in this case it clearly under-predicts the aerosol extinction along the whole vertical extension (NMB of about -60 %). This view, and particularly the steeper decrease of the aerosol extinction with height



in the lowermost levels, also points to some underestimation of the height of the PBL by the model. This aspect is however beside the scope of the present work.

### 4.3.1 Comparison to ground-level PM values

A further quantitative evaluation of the model ability to reproduce the observed aerosol/dust load is given in Figure 11, where

the aerosol mass predictions at the particular vertical level coincident with the ground are compared to in-situ measured $PM_{10}$ and $PM_{2.5}$ data. In particular, model-simulated (red curve) and measured (black crosses) $PM_{10}$ (Figures 11 a,c) and $PM_{2.5}$ (Figures 11b,d) data are shown for both hourly (top panels) and daily-resolved (bottom panels) values, the latter being the metrics regulated by the European Air Quality Directive 2008/50/EC (EC, 2008). The contribution of dust to the total $PM_{10}$ and $PM_{2.5}$ as estimated by the model is also shown in the plots (dust-$PM_{10}$ and dust-$PM_{2.5}$, red dashed lines). Overall, the

results show a good reproduction of the aerosol mass by the model in dust-free or low-dust conditions (i.e., before May 20 and on May 25), but also highlight a model overestimation of both $PM_{10}$ and $PM_{2.5}$ in desert dust conditions (May 20-24). The hourly-resolved temporal evolution of the $PM_{10}$ and $PM_{2.5}$ fields also reveals some 12h-time shift of the PM maximum values, these being observed around midday of May 22 and predicted in the night between 22 and 23 May by the model. This time-shift is however somehow hidden/modulated in the daily-average comparison, the latter showing a better synchronization

between the two, with coincident maxima on May 22. The model overestimation for the dust peak (up to 70 % in $PM_{10}$ and $PM_{2.5}$) is however obviously still evident on this metric, and evidently related to an overestimation of the dust component in terms of mass. For comparison, we report in Figure 11c (dashed lines, cross symbols) the contribution of desert dust to the $PM_{10}$ as estimated in the measurement site following the methodology reported by Barnaba et al. (2016). This is a modification of the 'standard' methodology suggested by the European Commission (EC, 2011), which was shown to overcome some of

the limitations of the latter, and to better fit data over Italy.

### 5 Conclusions

In this study, we evaluated the WRF-Chem model capability to reproduce the dust outbreak occurred in the period 19-24 May 2014 over the Central Mediterranean regions. The intrusion of mineral dust was caused by a synoptic situation in the

Mediterranean corresponding to an omega-like pressure configuration with a cyclogenesis in the Atlantic coasts of Spain. In fact, the cyclogenesis brought strong westerly Atlantic winds to the northern Sahara while the northward transport was made possible by the ridge associated with the omega-like pattern. In general, the synoptic conditions for the geopotential height at 700 hPa were well reproduced by WRF-Chem. This allowed us to simulate with good confidence the path of the dust during the northward intrusion. Concerning the synoptic conditions for the water vapor mixing ratio, although the simulated and

observed patterns are qualitatively similar, we found some differences in their intensity and extension, as the model generally overestimates the water vapor mixing ratio transport from Africa.

The overlapping between the modeled AOD parameter with the source emission functions allowed identifying the desert regions responsible for the investigated northward dust intrusion. Confirming some recent findings from Ginoux (2012), in



this case we recognized a persistently active source region of desert dust located south of the Atlas Mountains and between Algeria and Tunisia. This region is characterized by a system of ephemeral salt lakes that stays dry in summer, but receives some water in winter playing an important role in modulating dust emissions.

Using a multi-platform observational dataset of aerosol (and desert dust) properties, we tested the ability of the model to
reproduce the horizontal and vertical pattern of the dust intrusion, which was found to be composed by several, superimposed, time-shifted dust-pulses.

The comparison with the MODIS-AOD satellite retrievals showed the WRF-Chem simulation to satisfactorily resolve the arrival, temporal evolution and horizontal extent of the plumes over the Central Mediterranean. Results also showed a good agreement between the model-and the AERONET-AOD measured from the ground at selected Mediterranean sites. The
analysis of AERONET data and its comparison with simulations showed that the first dust intrusion occurred on 21 May reaching southern Sardinia, the second and more intense dust plume occurred on 22 May, penetrating up to northern Corsica and central Italy. This result confirms and integrates the analysis from polar satellite data, the latter being limited in time.

The characterization of the aerosol field over the vertical scale was made here by employing continuous lidar/ceilometer measurements performed in a single observational point that lies just in the middle of the area investigated (Rome, Italy), and
was therefore expected to be particularly suitable to evaluate the model capability to reproduce the dust plume vertical extent and its transport timing. Overall, the lidar measurements in the period 19-25 May clearly highlighted the 'pulsed' nature of the event discussed here. In most of the cases the desert dust is shown to arrive above the PBL and then to descend and mix with the local aerosols within it. The comparison with lidar measurements also highlighted that the good matching of the model and AERONET AOD comes from a good reproduction of the aerosol extinction coefficient along the profile (NMB of about 50%),
with the best performances in terms of aerosol optical properties during the maximum of the dust event. When the model-measurement comparison was done at ground-level in terms of aerosol mass ($PM_{2.5}$ and $PM_{10}$ data are used for this purpose), a tendency to overestimate the desert dust aerosol mass was conversely revealed. Such an overestimation reaches 70% for $PM_{10}$ and $PM_{2.5}$, respectively during the dust peak, reduced to 30 % and 10 – 60 % in weak-dust or no-dust conditions (before May 20 and after May 23). For $PM_{10}$, the comparison showed the mass overestimation to be driven by an overestimated dust
contribution of the order of 140%. This result points to a possible over-prediction of the number of large dust particles by the model (affecting dust mass more than optical properties).

**Acknowledgments**

Analyses and visualizations used in this paper were produced with the Giovanni online data system, developed and maintained
by the NASA GES DISC (http://giovanni.gsfc.nasa.gov/). We also acknowledge the MODIS and AIRS mission scientists and associated NASA personnel for the production of the data used in this research effort. We thank the principal investigators and their staff for establishing and maintaining the AERONET sites, the data from which have been used in this study. The aerosol

and desert dust observations in Rome (Italy) employed in this work were supported by the EC-LIFE+2010 DIAPASON project (LIFE+10 ENV/IT/391).

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





Figure 1: The numerical domain/ topography and the location of the AERONET stations used in this study;

Figure 2: Time Averaged Maps of Geopotential Height obtained merging the Nighttime/Descending and Daytime Ascending 1 deg. AIRS AIRX3STD v006 data over (a) 2014-05-16 - 2014-05-18 at 500 hPa; (b) 2014-05-19 - 2014-05-24 at 500 hPa; (c) 2014-05-16 - 2014-05-22 at 850 hPa; (d) 2014-05-23 - 2014-05-24 at 850 hPa;

Figure 3: Geopotential Height 1 deg @700hPa [AIRS AIRX3STD v006] for May 21 (a), May 22 (c) and May 23 (e) ; WRF-Chem Geopotential Height and wind fields (u,v) at 700 hPa for May 21 (b), May 22 (d) and May 23 (f);

Figure 4: Water vapor mixing ratio 1 deg @700hPa [AIRS AIRX3STD v006] for May 21 (a), May 22 (c) and May 23 (e) ; WRF-Chem qvapor at 700 hPa for May 21 (b), May 22 (d) and May 23 (f);

Figure 5: Model-AOD (shaded contour) and the dust source strength (white contour), black arrows (u,v) at 1000 hPa, and white arrows (u,v) at 700 hPa. The oval-dotted contours (S1, S2, S3, S4) denote the locations of the desert regions responsible for the northward dust intrusion;

Figure 6: MODIS-Terra Combined Dark Target and Deep Blue AOD at 550 nm for land and ocean at 1 deg-resolution as retrieved for: (a) May 20, (b) May 21, (c) May 22, (d) May 23;

Figure 7: The model-AOD distribution (shaded contour) and the wind field components (u,v) at 700 hPa (black arrows) for (a) May 20, (b) May 21, (c) May 22, (d) May 23;

Figure 8
Hourly-resolved columnar AOD at 550 nm from WRF-Chem simulation (continuous red line) and measurements (black squares) at the AERONET stations of (a) Carloforte, (b) Rome, (c) Ersa, (d) Modena, (e) Calern and (f) Ispra;

Figure 9: Time-height plot at Rome of (a) the CHM15K ceilometer range corrected (RC) signal (S) (RCS = $\ln(SxR^2)$, (b) of the ATLAS lidar volume depolarization (%), and (c) of the model total dust mass (µg/kg dry air);

Figure 10: Vertical profile (0- 5 km, Y axis) of the 1-hour mean aerosol extinction coefficient (1/km) in Rome as retrieved by the ATLAS lidar (at 532 nm, orange diamonds, red line) and simulated by the WRF-Chem model (at 550 nm, gray squares, black line). Error bars associated to the lidar data represent an expected 30% error in the lidar retrieval (see text), while a 50% error bar has been associated to the model values. Top (a, b) and bottom (c, d) panels refer to May 22 and May 25, respectively. Left (a, c) and right (b, d) column panels to profiles at 0:30 and 4:30 UTC, respectively.

Figure 11: (a) Hourly-resolved evolution of modeled $PM_{10}$ (red continuous line) and dust-$PM_{10}$ (red dotted line), and relevant hourly (black crosses) and 3-hour-running average (blue continuous line) experimental data; (b) same as (a) but for $PM_{2.5}$; (c) daily-resolved evolution of modeled $PM_{10}$ (red continuous line with star points) and modeled dust-$PM_{10}$ (red dotted line with star points), and relevant daily-average $PM_{10}$ from measurements (blue continuous line with cross points), and daily-average dust-$PM_{10}$ estimated from measurements (blue dotted line with star points); (d) same as (c) but for $PM_{2.5}$; no dust-$PM_{2.5}$ estimated from measurements was available in this case.





| | option number | namelist variable | Model |
|---|---|---|---|
| Land surface | 3 | sf_surface_physics | RUC model |
| PBL model | 5 | bl_pbl_physics | MYNN 2.5 level |
| Surface similarity | 1 | sf_sfclay_physics | MM5 Similarity Scheme |
| Microphysics | 2 | mp_physics | Purdue Lin |
| Short-wave radiation | 4 | ra_sw_physics | RRTMG |
| Long-wave radiation | 4 | ra_lw_physics | RRTMG |
| Aerosol mixing rules | 2 | aer_op_opt | Maxwell-Garnett |

Table 1: namelist settings of the physical parameterizations

| Namelist variable | Opt_number | registry.chem package |
|---|---|---|
| chem_opt | 300 | GOCART_SIMPLE |
| dust_scheme | 1 | Shao_2001 |
| dust_opt | 4 | *dustuoc* |

Table 2: namelist settings of the chemical and dust emission parameterization





Figure 1

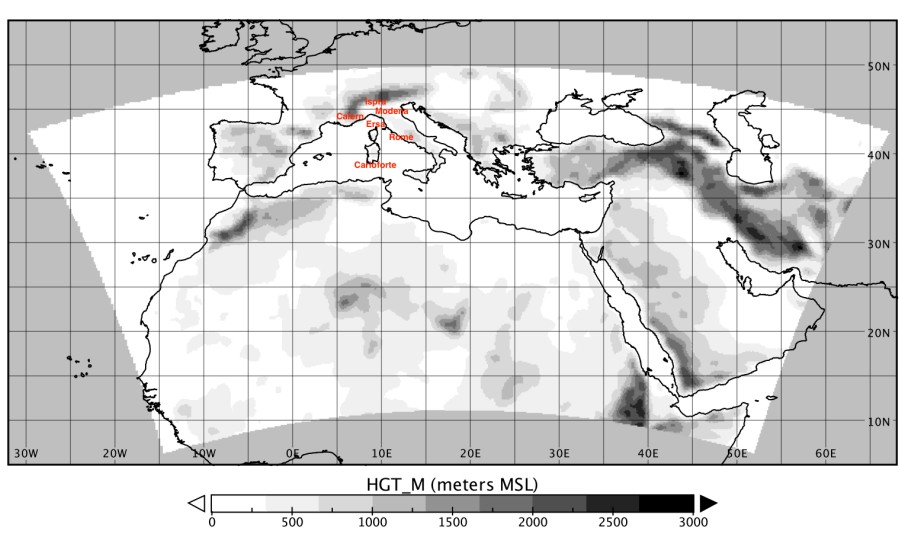

Figure 2

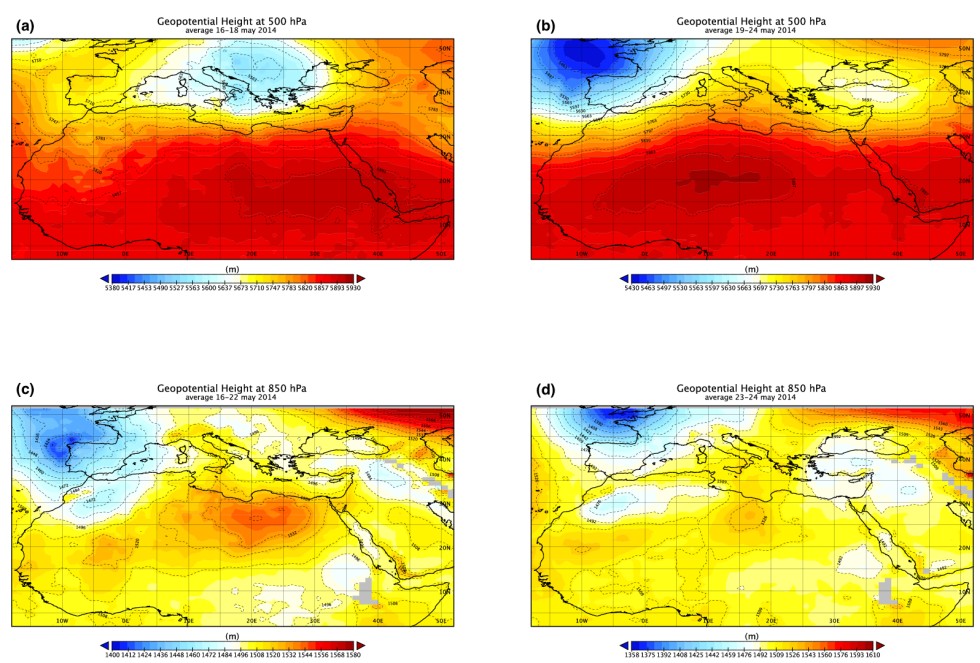





*Figure 3*

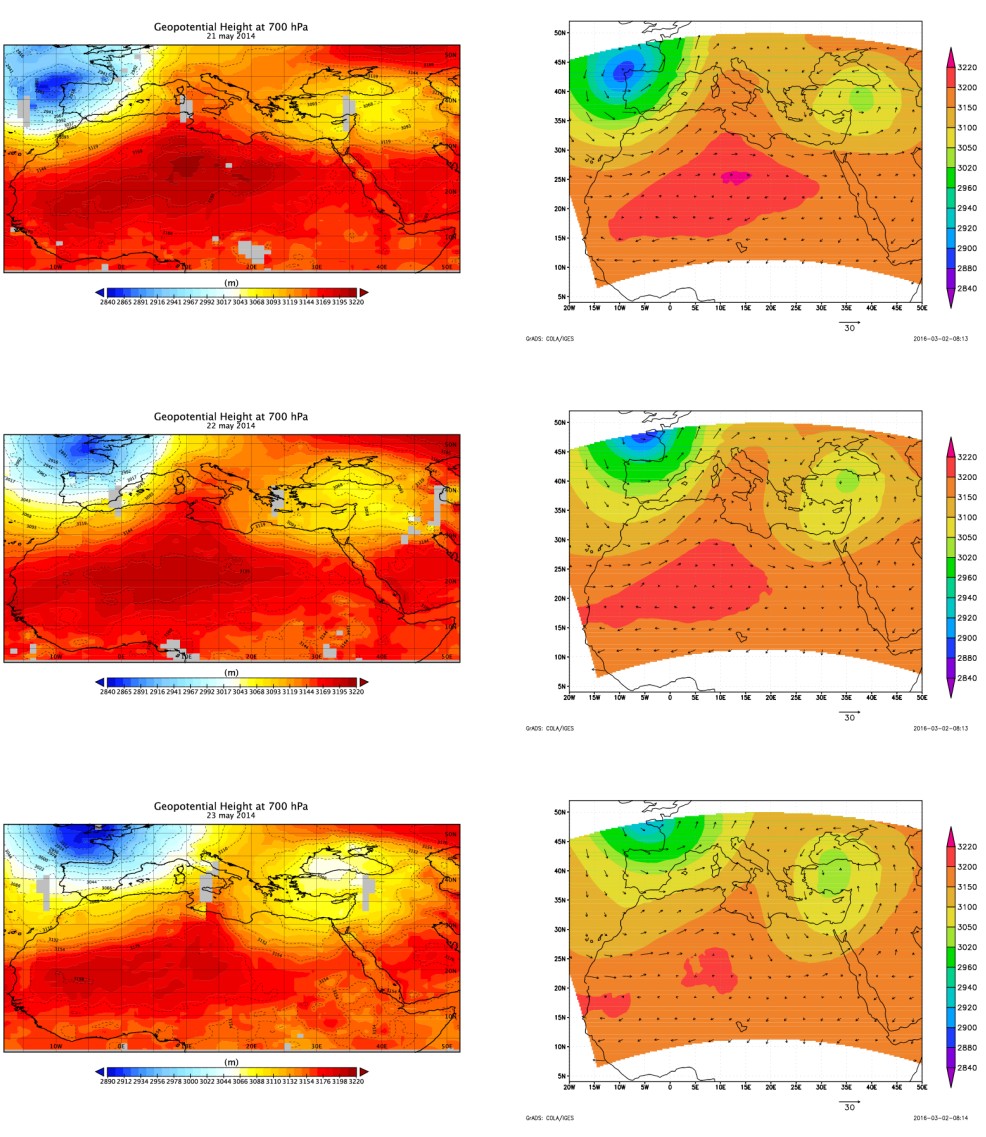





*Figure 4*

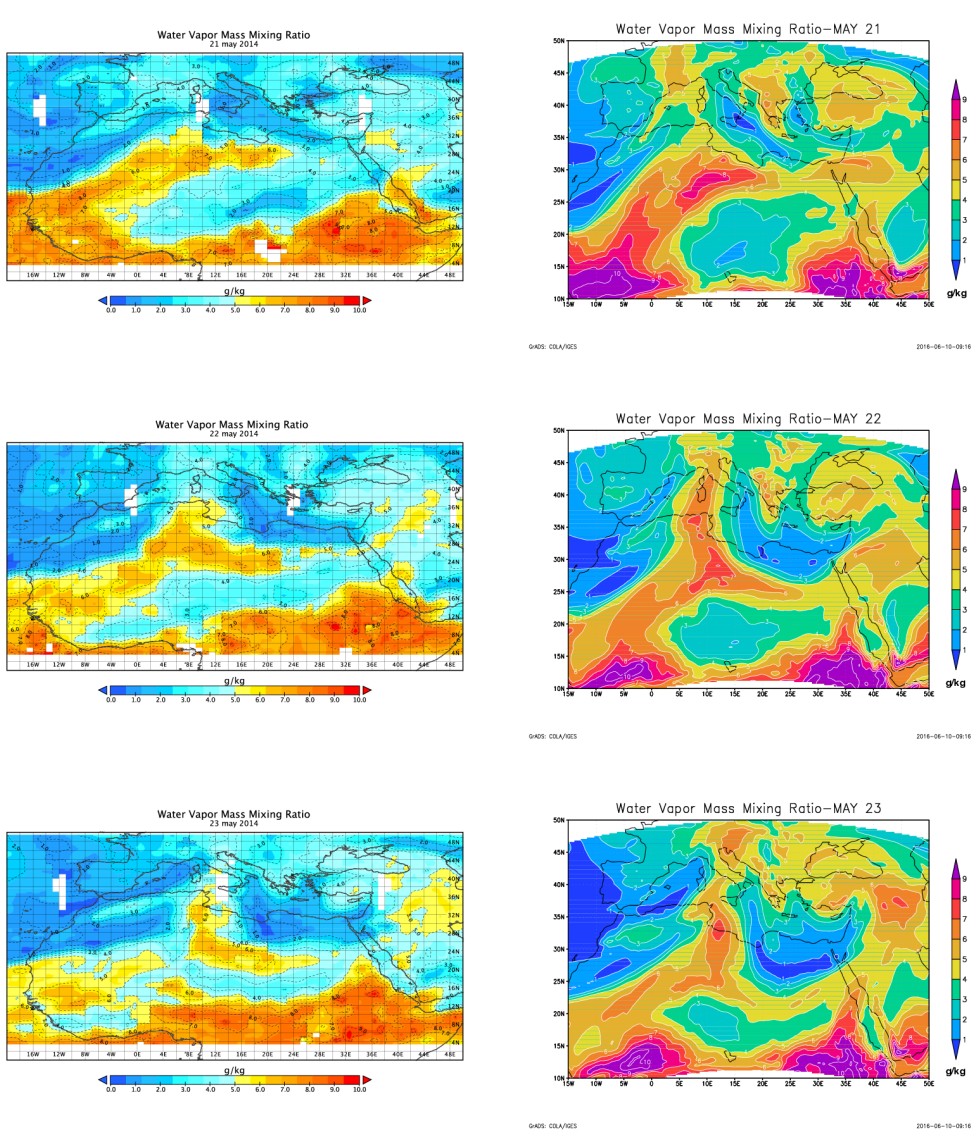





*Figure 5*

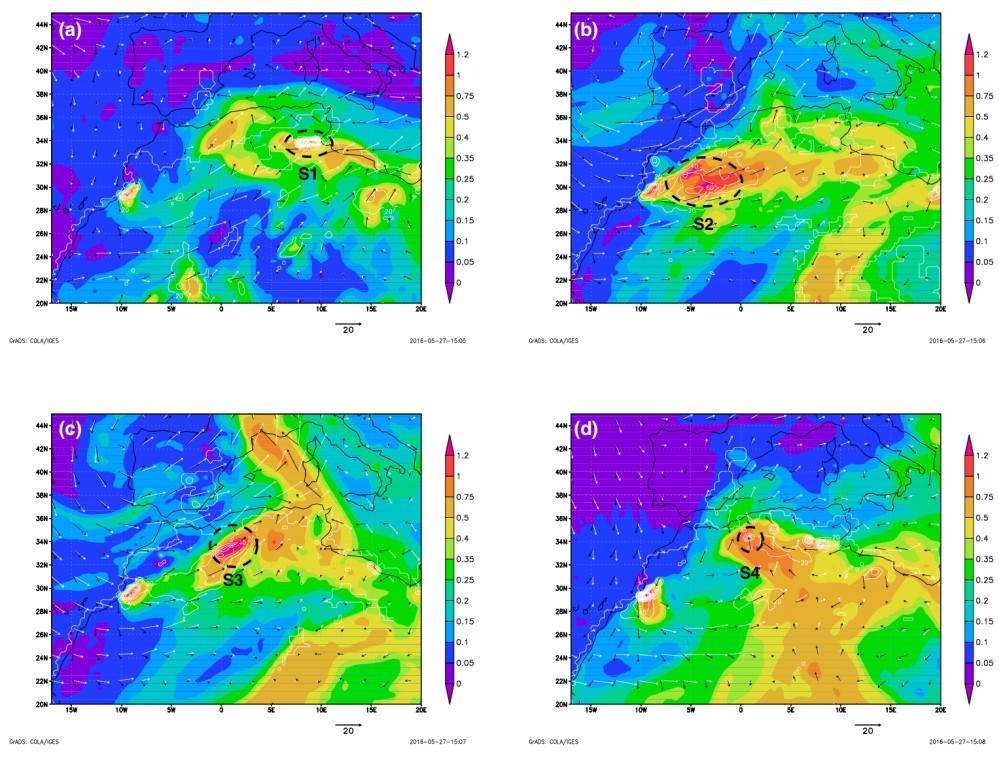



*Figure 6*

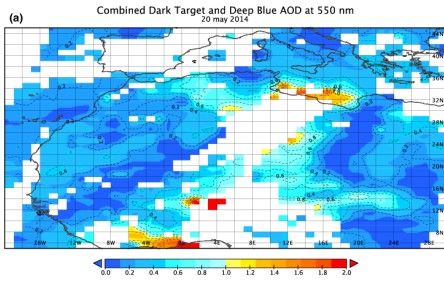
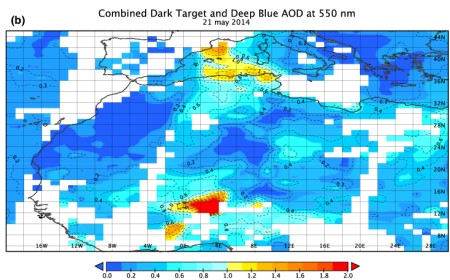

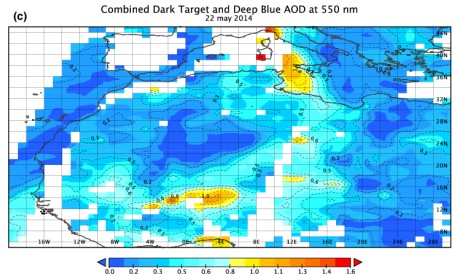
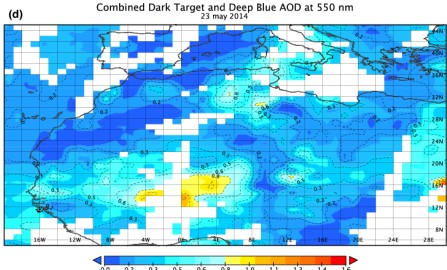




*Figure 7*

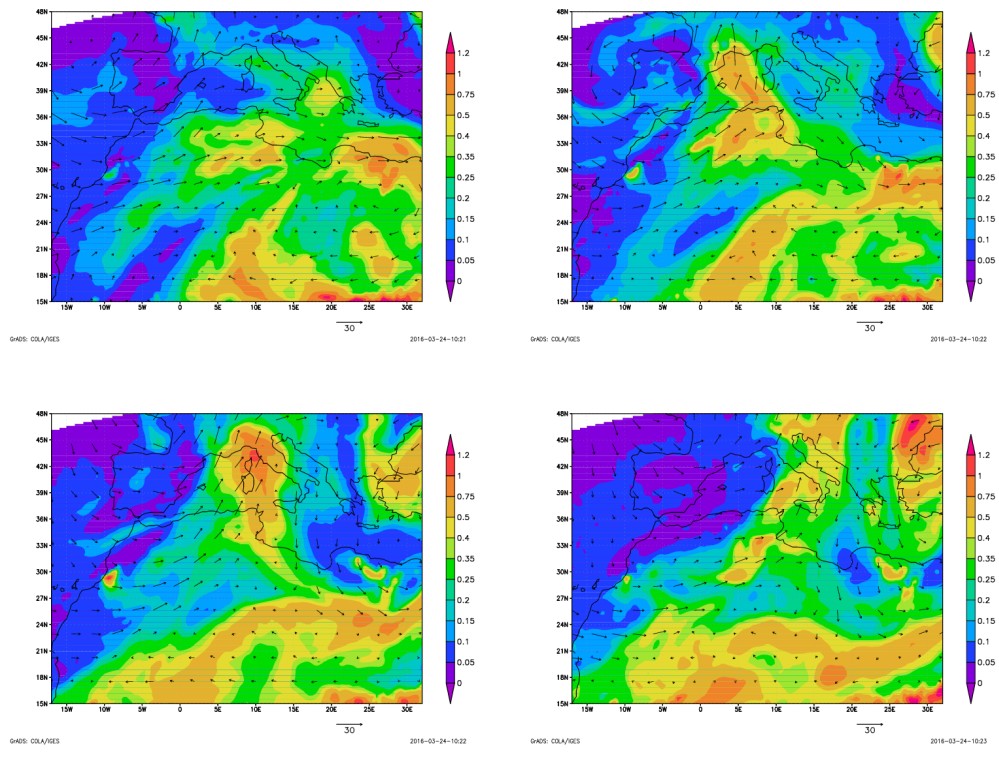



*Figure 8*

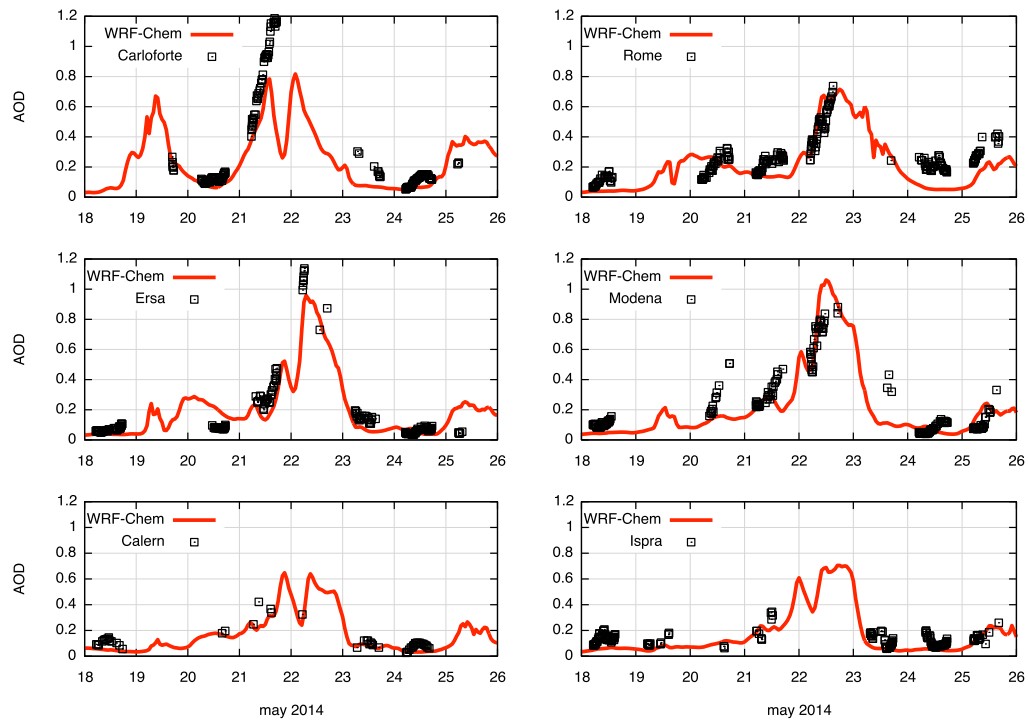





*Figure 9*

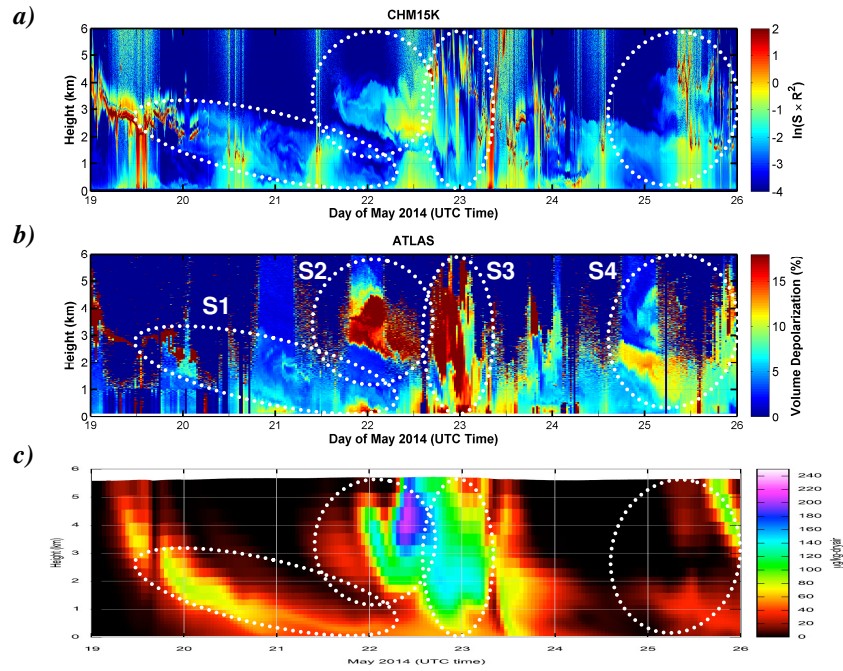






*Figure 10*

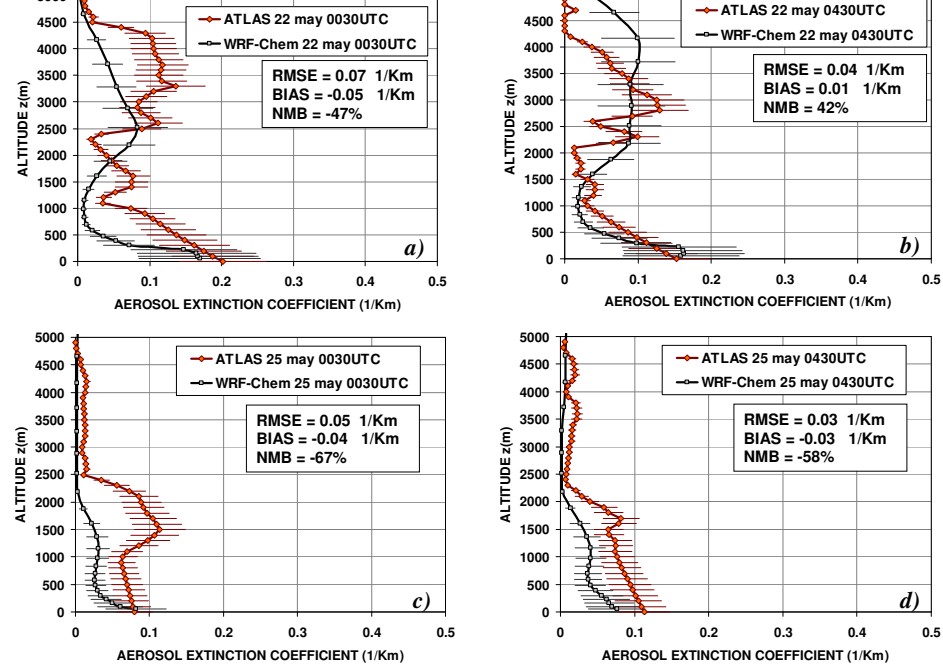



*Figure 11*

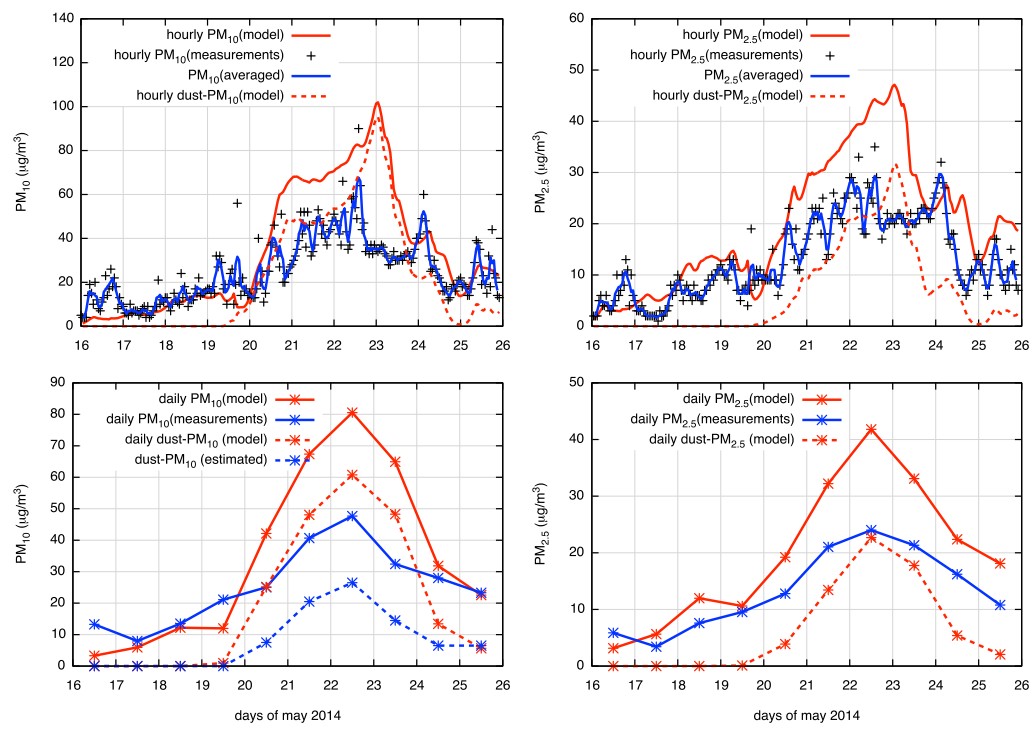