# Peer review of "WRF-Chem model simulations of a dust outbreak over the Central Mediterranean and comparison with multi-sensor desert dust observations"

_Atmospheric Chemistry and Physics, 2016_

## Referee Comment (RC1) · Anonymous Referee #1 · 12 Sep 2016

In this paper, the WRF-Chem was used to simulate an intense Saharan dust outbreak event that took place over the Mediterranean in May 2014. Results have shown that a cyclone near the Atlantic coasts of Spain is responsible for strong westerly Atlantic winds (about 20 m s-1) reaching the northern Sahara and leading to the lifting of mineral dust. The northward transport is made possible by a ridge over the central Mediterranean associated with the omega-like pressure configuration. Compared with optical properties from satellite and ground-based sun-photometers and lidars, plus in situ PM10 data, the WRF-Chem data showed a good agreement with them in different aspects.

In general, the comparison between WRF-Chem and other multi-sensor desert dust

observations maybe a good point. However, the manuscript needs to be extensively improved in some details. I strongly advice the authors to take into consideration of the following minor remarks so as to improve the quality this manuscript.

Comments

1. The abstract is too long and need to be simplified so that the readers can catch the major points and results.

2. This paper doesn't have key words, please add them.

3. I would suggest authors include more recent paper in this field to strengthen the introduction section. The following paper is for reference only:

(1) Shao, Y., et al., 2011: Dust cycle: An emerging core theme in Earth system science. Aeolian Research, 2.4 (2011): 181-204.

(2) Huang, J., T. Wang, W. Wang, Z. Li, and H. Yan, 2014: Climate effects of dust aerosols over East Asian arid and semiarid regions, Journal of Geophysical Research: Atmospheres, 119, 11398–11416, doi: 10.1002/2014JD021796.

(3) Wang, W. et al., 2010: Dusty cloud properties and radiative forcing over dust source and downwind regions derived from A-Train data during the Pacific Dust Experiment, Journal of Geophysical Research, 115, D00H35, doi:10.1029/2010JD014109.

(4) Chen, S., et al., 2013: Modeling the transport and radiative forcing of Taklimakan dust over the Tibetan Plateau: A case study in the summer of 2006, Journal of Geophysical Research: Atmospheres, 118 , doi:10.1002/jgrd.50122.

(5) Bi, J.et al., 2011: Toward characterization of the aerosol optical properties over Loess Plateau of Northwestern China, Journal of Quantitative Spectroscopy  Radiative Transfer, 112 (2), 346-360.,doi:10.1029/2009JD013372.

4. Page 10 As we known, many factors such as Wind speed, Atmospheric stability, and so on play an important role in dust emission, why are the two factors more important?

Why do you show the Fig.4 in the paper?

5. Page10, Line 32 Is the threshold calculated in this paper or obtain from other literatures? This paper did not tell us explicitly.

6. Page11 "the total dust flux (white contours for the selected dates of May 18, 20, 21 and 24 (panels a, b, c, and d respectively). The AOD is obtained from WRF-Chem simulations vertically integrating (from the ground to the top of domain, i.e. 20 km) the aerosol extinction coefficient at 550 nm. The same figures also show: i) the wind field at 10 m (black arrows), that is directly connected with the dust emission, and 5 ii)" Please check the brackets whether match or not.

7. Page11, Line32 What's a system of ephemeral salt lakes effect on the four dust sources?

8. Page15, Line16-17 What's the reasons that the model overestimated the dust peak (PM2.5 and PM10 )?

9. Fig. 1: The fonts in the map are too small that they are difficult to read.

10. Fig.2: You can use the same color bar in Fig.2.

11. Fig.3 and Fig.4: You should use the same domain, map projection, and color bar.

12. Fig. 11: The figure seems to be very busy. Could you modify it?

---

## Referee Comment (RC2) · Anonymous Referee #2 · 30 Sep 2016

General Comments

This study evaluated the WRF-Chem simulation for a dust event over the Central Mediterranean in May 2014. The evaluation used multiple observations including satellite and ground data. Understanding the dust emissions over Sahara and how those dust particles can be transported towards the Mediterranean is an important topic. However, the evaluation and analysis presented in this study are in poor quality. Among the comparisons between the model and observations, I can only see the good comparison with AERONET AOD. All other comparisons are not good. However, the authors did not spend efforts to improve the simulation or conduct any sensitivity experiment to provide any suggestions to improve the model against observations in

future. This makes the study less significant. More specific comments listed below.

Specific Comments

1. In the model setup session, please describe how long does the simulation last? What's the initial date and chemical initial and boundary condition?

2. Figure 2 shows geopotential distribution from the AIRX3STD not from a reanalysis. However, later on, many discussion about the wind fields. Then, why not just using a reanalysis dataset for both geopotential and winds? Please explain.

3. Line 5-9 of page 10, the authors evaluated water vapor mixing ratio and claimed that water vapor is important for chemistry. However, do we expect significant impact from chemistry in this study? If yes, is GOCART-simple scheme too simple for complex chemistry involving dust? Don't see the reason for evaluating water mixing ratio for this study.

4. Figure 3 and 4, when comparing model results and measurements, they need to be shown in the same map projection and domain. The current format is very confusing. The label showed blow each figure indicates "GRADS" and "date" needs to be removed.

5. Figure 5 is too busy. Suggest separating AOD and emission. Emission color contour with 10-m winds, and AOD color contour with 700 hPa winds.

6. Again, Figure 6 and 7 need to be on the same map project and domain for direct comparison. The current format is too confusing. The two figures are also with different color tables. It seems to me that there is significant difference between MODIS and model. Is it due to the different map projections and color tables? What is the spatial correlation coefficient between the measurements and simulations?

7. Figure 8, comparing with AERONET, please show the mean bias and temporal correlation coefficient for each site.

8. Quality of Figure 9 needs to be improved.

9. Figure 10, I didn't see that the model reproduces vertical structures. Please show the vertical correlation coefficient for each profile.

---

## Author Comment (AC1) · 24 Nov 2016

**Authors Reply to Referee 1 (referee's comments in blue italics and our response in black)**

*In this paper, the WRF-Chem was used to simulate an intense Saharan dust outbreak event that took place over the Mediterranean in May 2014. Results have shown that a cyclone near the Atlantic coasts of Spain is responsible for strong westerly Atlantic winds (about 20 m s-1) reaching the northern Sahara and leading to the lifting of mineral dust. The northward transport is made possible by a ridge over the central Mediterranean associated with the omega-like pressure configuration. Compared with optical properties from satellite and ground-based sun-photometers and lidars, plus in situ PM10 data, the WRF-Chem data showed a good agreement with them in different aspects. In general, the comparison between WRF-Chem and other multi-sensor desert dust observations maybe a good point. However, the manuscript needs to be extensively improved in some details. I strongly advice the authors to take into consideration of the following minor remarks so as to improve the quality this manuscript.*

We would like to thank the Referee1 for the useful and valuable comments in his/her report. Point-by-point responses are included below. We would also like to highlight that some main modifications to the manuscript have been introduced following the Referee2 suggestions. In particular, in order to show the advantages of using a physics-based dust emission scheme (Shao, 2001, now S01 in the text), we added the corresponding results using the simplified emission scheme by Shao, 2011 (S11 in the text). This can be considered a 'minimal' version (in term of internal parameters) of the S01 emission scheme used in this study. Not to change the original structure of the paper, this additional results have however been included into a separate Appendix, and commented within the main text where appropriate.

*Comments*
*1. The abstract is too long and need to be simplified so that the readers can catch the major points and results.*
We tried to shorten and simplify the abstract, also introducing some of the modifications inserted in the revised version.

*2. This paper doesn't have key words, please add them.*
To our knowledge, no key word is required by ACP and therefore these were not provided. If necessary, the following key words can be associated to the present study: Desert Dust modeling; Desert dust observations; WRF-Chem simulations; Mediterranean dust outbreak; Saharan dust emission

*3. I would suggest authors include more recent paper in this field to strengthen the introduction section. The following paper is for reference only:*
*(1) Shao, Y., et al., 2011: Dust cycle: An emerging core theme in Earth system science. Aeolian Research, 2.4 (2011): 181-*

*204.*

*(2) Huang, J., T. Wang, W. Wang, Z. Li, and H. Yan, 2014: Climate effects of dust aerosols over East Asian arid and semiarid regions, Journal of Geophysical Research: Atmospheres, 119, 11398–11416, doi: 10.1002/2014JD021796.*

*(3) Wang, W. et al., 2010: Dusty cloud properties and radiative forcing over dust source and downwind regions derived from A-Train data during the Pacific Dust Experiment, Journal of Geophysical Research, 115, D00H35, doi:10.1029/2010JD014109.*

*(4) Chen, S., et al., 2013: Modeling the transport and radiative forcing of Taklimakan dust over the Tibetan Plateau: A case study in the summer of 2006, Journal of Geo- physical Research: Atmospheres, 118 , doi:10.1002/jgrd.50122.*

*(5) Bi, J.et al., 2011: Toward characterization of the aerosol optical properties over Loess Plateau of Northwestern China, Journal of Quantitative Spectroscopy Radiative Transfer, 112 (2), 346-360.,doi:10.1029/2009JD013372.*

We thank Referee1 for his/her suggestions, we added reference to those papers in the text.

*4. Page 10 As we know, many factors such as Wind speed, Atmospheric stability, and so on play an important role in dust emission, why are the two factors more important? Why do you show the Fig.4 in the paper?*

We originally included water vapor as this is a key parameter driving the horizontal AOD field investigated in the manuscript. We agree that it is not the only one and, as this question was also raised from Referee2, we understood this point was neither clear nor exhaustive, therefore, following the revision process, we decided to eliminate Fig. 4 and the relevant comments from the manuscript.

*5. Page10, Line 32 Is the threshold calculated in this paper or obtain from other literatures? This paper did not tell us explicitly.*

The range of values reported in the text is taken from the experimental campaign conducted by Chomette et al., (1999). It was conducted in seven selected sites in the Saharan and Sahelian deserts. This missing information is now added to the text (Section 4.2.1). We also updated the reference Chomette et al., (2006) with  Chomette et al., (1999).

*6. Page11 "the total dust flux (white contours for the selected dates of May 18, 20, 21 and 24, (panels a, b, c, and d respectively). The AOD is obtained from WRF-Chem simulations vertically integrating (from the ground to the top of domain, i.e. 20 km) the aerosol extinction coefficient at 550 nm. The same figures also show: i) the wind field at 10 m (black arrows), that is directly connected with the dust emission, and 5 ii)" Please check the brackets whether match or not.*

We thank the Referee for noting this typo. In the final version we changed the Figure the text above is referring to (old Figure 6, now Figure 4). However, we took care in correctly using the brackets. The relevant text now reads: "… and the total dust flux calculated with the S01 scheme (black contours for the selected dates of May 18, 20, 21 and 24; panels a, b, c, and d respectively)." (Section 4.2.1)

*7. Page11, Line32 What's a system of ephemeral salt lakes effect on the four dust sources?*

As it is explained in the text, these ephemeral lakes or "Chotts" are located in a large region south of Atlas Mountains that

stays dry during spring-summers period, thus constituting an important source production area for dust outbreaks in the Mediterranean. This was first revealed by Ginoux et al. (2012). In particular, as can be easily verified in their figure 7, the white contours denoted as N.19-20-21-22, indicate the "Chotts" region which overlap our source regions (with the S01 emission model) and reported as S1,S2,S3,S4 in our figure 4.

*8. Page15, Line16-17 What's the reasons that the model overestimated the dust peak (PM2.5 and PM10 )?*

We believe this is due to two main factors: 1) over-prediction of large particles by the model, and 2) a missing wet deposition process within the model.

Point 1 is because we do not note such a high overestimation in terms of aerosol extinction (e.g. old Figure 10, now Figure 8). This indicates that the model likely over-predicts those particles having the highest mass but moderate to negligible aerosol extinction.

Point 2 is because, even if we introduced a wet deposition scheme within the model (Section 2.1.2), it only considers non-convective precipitation as active in the wet removal. In fact the scheme of Balkanski et al (1993) we used and extended to desert dust, only use large-scale precipitations (non convective), that is about the 60% of the total. A more complete WRF-Chem wet deposition scheme for dust aerosols coupled with the aerosols/chemistry GOCART mechanisms has been only released starting from version 3.8, but it is not fully available to the community yet.

We inserted a comment in the text on these aspects (Section 2.1.2)

*9. Fig. 1: The fonts in the map are too small that they are difficult to read.*

*10. Fig.2: You can use the same color bar in Fig.2.*

*11. Fig.3 and Fig.4: You should use the same domain, map projection, and color bar.*

*12. Fig. 11: The figure seems to be very busy. Could you modify it?*

Comments 9-12 have been all addressed in the revised version: a) fonts have been enlarged, b) the same color bar is now used when representing the same variable as derived from model or measurements, c) the same map projection for both model and measurements is now used, d) the old Figure 11 (now Figure 9) has been simplified.

**Authors Reply to Referee 2* (referee's comments in blue italics and our response in black)**

*This study evaluated the WRF-Chem simulation for a dust event over the Central Mediterranean in May 2014. The evaluation used multiple observations including satellite and ground data. Understanding the dust emissions over Sahara and how those dust particles can be transported towards the Mediterranean is an important topic. However, the evaluation and analysis presented in this study are in poor quality. Among the comparisons between the model and observations, I can only see the good comparison with AERONET AOD. All other comparisons are not good. However, the*

*authors did not spend efforts to improve the simulation or conduct any sensitivity experiment to provide any suggestions to improve the model against observations in future. This makes the study less significant. More specific comments listed below*

We thank the referee for the useful and valuable comments in his/her report. A reply to his/her general comments is reported hereafter, followed by point-by-point responses to the his/her specific comments.

We would however also mention at first that we do not fully agree with Referee2 on the fact that the model only shows good performances in the comparison with AERONET AOD. Indeed the comparison with the AERONET AOD is good (and, based on his/her comment, in the revised version we further quantify this through statistical parameters in a new Table 3). Nevertheless this 'good' model-derived AOD at specific sites (i.e., within specific model cells), is exactly the same used in the comparison with the satellite one (MODIS instrument) over a broader area. In fact, our view is that this broader (geographically extended) comparison of the WRF-Chem AOD with the MODIS one shows a very nice agreement between the two. However, thanks to the Referee2 comments, this comparison is certainly facilitated in this new version (improved relevant Figures, see also point-by-point replies below).

The comparison with lidar is also relatively good and model-vs.-measurements discrepancies are similar to those typically found in lidar-vs.-model comparisons studies (e.g., Mona et al., 2014). To quantify the lidar-WRF-Chem agreement, in addition to the statistical metrics already introduced in the first version, we further provide now the correlation coefficients (R) of the measured-vs-model profiles, as suggested by the Referee (new Figure 8, upgrading the old Figure 10). These show a moderate (R = 0.6) to excellent (R = 1.0) correlation between the modeled and measured aerosol extinction profiles, although some important differences exist, as already pointed out in the first version. This comparison was (and is) intended to better disclose the reasons for agreement/disagreement of the AOD values previously discussed. In fact, being the AOD defined as the integral over altitude of the aerosol extinction, this view allows to understand for example if the 'good' model-AOD comes from 'compensation effects' between aerosol extinction underestimations/overestimations at different altitudes or not, which could indicate poor reproduction of the vertically-resolved desert dust transport. Our results (old Figure 10, now Figure 8) suggest that this is generally not the case, and the model rather well captures the main desert dust stratifications within the column, although with a general underestimation in the lowermost levels (< 1500 m).

Linked to that, and in addition to these considerations, we would also like to emphasize that when comparing models to observations even a 'bad result' (poor agreement) is 'a result'. In fact, this indicates that more efforts should be done in the direction of improving the 'bad' result, rather than in the direction of 'refining' the already 'good' ones (or in parallel to).

This is the case of the modeled PM2.5 and PM10 fields, which in the investigated case are clearly badly reproduced by the model (e.g., old Figure 11, now Figure 9). If it is true that, as mentioned above, the model-lidar comparison highlighted a clear tendency to under-predict the aerosol field below 1500 m altitude, but this translates into an even more evident under-estimation of PM, meaning that extinction-to-PM conversion (related to the assumed particle size) is clearly badly

reproduced. On this aspect, we now also speculate in the text that this 'bad' result is (at least partially) related to some missing wet deposition process within the model. In fact, even if we introduced the Balkanski et al (1993) wet deposition scheme within the model (Section 2.1.2) and extended it to desert-dust particles, this only considers large-scale precipitation (and not convective precipitation) as active in the wet removal. In our case we verified that, according to the simulation, convective precipitation over the Central Mediterranean during the dust outbreak reaches up to 50% of the total precipitation. Starting from the Referee2 criticism, specific comments on all these aspects have now been included in the new text (Section 4.3.1), also mentioning that, as proposed, future work should be devoted to better disclose the reasons for model vs. measurements mismatches, particularly on the PM metrics. In this respect also note that the complete WRF-Chem wet deposition scheme for dust aerosols coupled with the aerosols/chemistry GOCART mechanisms has been released only very recently (only starting from WRF-Chem version 3.8), and it is not fully available to the community yet.

As a second important point raised by Referee2, we agree that some 'sensitivity tests' were missing in the first version of the manuscript. This was partially due to the fact that a preliminary study (Rizza et al., 2016) was already dedicated to the matter. In fact, that study showed that the emission scheme actually implemented within WRF-Chem 3.6.1, under the package "GOCART-AFWA", produces a marked over-prediction of dust emissions and a consequent overestimation of the dust concentration over the Mediterranean. In the revised version of this manuscript we now provide more details on those early results, in order to better highlight the reason for choosing to test a more complete physics-based emission scheme (that by Shao et al., 2001) within this study. However, starting from the Referee criticism on this point, in the revised version we now further test the model sensitivity to the dust emission scheme, introducing a second simulation which makes use of the Shao (2011) emission scheme (refereed to as S11 in the manuscript). This can be considered a 'minimal' version (in terms of internal parameters) of the Shao (2001) physical-based emission scheme (now referred to as S01) used in this study. Not to change the original structure of the manuscript, this additional results have been included into a separate Appendix and commented within the main text where appropriate. The reader is referred to that material to understand the benefits of the S01 scheme originally used. In fact, a main goal of the current study was (and is) to demonstrate that a physical-based emission scheme may be used with confidence in a regional/continental dust transport model. This is already a not trivial question, as evidenced by Shao et al., (2011b). Certainly, as mentioned, our future work will be devoted to the tuning of the several internal parameters that characterize this kind of size-resolved dust fluxes.

Specific Comments

*1. In the model setup session, please describe how long does the simulation last? What's the initial date and chemical initial and boundary condition?*

We thank the Reviewer2 for noting this missing information. The simulation lasted 10 days, starting at May 16 00UTC. An idealized vertical profile for each chemical species is provided to start the model simulation. This vertical profile is based

upon northern hemispheric, mid-latitude, clean environment conditions. Boundary conditions are obtained using the same methodology. On the other side the numerical domain has been chosen large enough to include all possible dust source, that in our case are localized in the Sahara desert. This has been added to the text in the revised Section 2.1.

*2. Figure 2 shows geopotential distribution from the AIRX3STD not from a reanalysis. However, later on, many discussion about the wind fields. Then, why not just using a reanalysis dataset for both geopotential and winds? Please explain.*

We accepted the Referre2 objection and, following his/her suggestion used reanalysis data from NCEP/NCAR (new Figures 2 and 3)

*3. Line 5-9 of page 10, the authors evaluated water vapor mixing ratio and claimed that water vapor is important for chemistry. However, do we expect significant impact from chemistry in this study? If yes, is GOCART-simple scheme too simple for complex chemistry involving dust? Don't see the reason for evaluating water mixing ratio for this study.*

We originally included water vapor as this is a key parameter driving the horizontal AOD field investigated in the manuscript. However, the objection of the Referee2 is correct and we understood this point was neither clear nor exhaustive, therefore, following the revision process, we decided to eliminate Fig. 4 and the relevant comments from the manuscript.

*4. Figure 3 and 4, when comparing model results and measurements, they need to be shown in the same map projection and domain. The current format is very confusing. The label showed blow each figure indicates "GRADS" and "date" needs to be removed.*

We thank Referee2 for this hint that allowed us to enhance the Figure readability. The new Figure 3 has now the same color palette, map projection and domain. Any label was removed. As described above the (old) Fig.4 has been removed.

*5. Figure 5 is too busy. Suggest separating AOD and emission. Emission color contour with 10-m winds, and AOD color contour with 700 hPa winds.*

We still prefer to merge the AOD and emission fields to better highlight the desert dust source areas. However, following this Referee2 comment, we tried to enhance the (old) Fig.5 readability (now Figure 4) by only plotting surface wind field (black arrow), AOD (shaded) and dust emission isolines (black contour).

*6. Again, Figure 6 and 7 need to be on the same map project and domain for direct comparison. The current format is too confusing. The two figures are also with different color tables. It seems to me that there is significant difference between*

*MODIS and model. Is it due to the different map projections and color tables? What is the spatial correlation coefficient between the measurements and simulations?*

Following this comment and to improve the readability of the information within the mentioned Figures we re-organized the content within the (old) Figures 6 and 7. In particular, in the revised version these have been merged into a single, multi-panel Figure (new Figure 5), each panel using the same color palette, map projection and domain extension.

As mentioned above in our reply to the Referee2 'General Comment', in our opinion this Figure shows a good reproduction of the AOD spatial pattern, especially during 21/22/23 of May. Indeed, with the modifications suggested a direct comparison is easier now. Furthermore, this same Figure but obtained with the S11 emission scheme has been included in the Appendix (Figure A2) and relevant comments have been reported in the relevant Section (4.2.2).

*7. Figure 8, comparing with AERONET, please show the mean bias and temporal correlation coefficient for each site.*

We added in a (new) Table 3 the requested mean bias and correlation coefficients.

*8. Quality of Figure 9 needs to be improved.*

Quality of (old) Figure 9 (new Figure 7) has been improved.

*9. Figure 10, I didn't see that the model reproduces vertical structures. Please show the vertical correlation coefficient for each profile.*

As suggested, the correlation coefficient R was added to the graphs (old Figure 10, now Figure8), to complement the other metrics already included in the original version.

**References:**

Mona, L., Papagiannopoulos, N., Basart, S., Baldasano, J., Binietoglou, I., Cornacchia, C., and Pappalardo, G.: EARLINET dust observations vs. BSC-BSC-DREAM8B8b modeled profiles: 12-year-long systematic comparison at Potenza, Italy, Atmos. Chem. Phys., 14, 8781–8793, doi:10.5194/acp-14-8781-2014, 2014.